# Effects of Straw Returning and New Fertilizer Substitution on Rice Growth, Yield, and Soil Properties in the Chaohu Lake Region of China

**DOI:** 10.3390/plants13030444

**Published:** 2024-02-02

**Authors:** Mei Luo, Ying Liu, Jing Li, Tingfeng Gao, Sheng Wu, Lei Wu, Xijun Lai, Hongjun Xu, Hongxiang Hu, Youhua Ma

**Affiliations:** 1College of Resources and Environment, Anhui Agricultural University, Hefei 230036, China; meiluo@stu.ahau.edu.cn (M.L.);; 2Nanjing Institute of Geography and Limnology, Chinese Academy of Sciences, Nanjing 210008, China; 3Station of Agricultural Environment Protection, Chaohu 238006, China

**Keywords:** straw returning, slow-release fertilizer, water-soluble fertilizer, rice growth, yield, soil chemical properties, enzyme activities

## Abstract

Recently, replacing chemical fertilizers with straw returning and new fertilizers has received considerable attention in the agricultural sector, as it is believed to increase rice yield and improve soil properties. However, less is known about rice growth and soil properties in paddy fields with the addition of different fertilizers. Thus, in this paper, we investigated the effects of different fertilizer treatments, including no fertilization (CK), optimized fertilization based on the medium yield recommended fertilizer amount (OF), 4.50 Mg ha^−1^ straw returning with chemical fertilizers (SF), 0.59 Mg ha^−1^ slow-release fertilizer with chemical fertilizers (SRF), and 0.60 Mg ha^−1^ water-soluble fertilizer with chemical fertilizers (WSF), on rice growth, yield, and soil properties through a field experiment. The results show that compared with the OF treatment, the new SF, SRF, and WSF treatments increased plant height, main root length, tiller number, leaf area index, chlorophyll content, and aboveground dry weight. The SF, SRF, and WSF treatments improved rice grain yield by 30.65–32.51% and 0.24–1.66% compared to the CK and OF treatments, respectively. The SRF treatment increased nitrogen (N) and phosphorus (P) uptake by 18.78% and 28.68%, the harvest indexes of N and P by 1.75% and 0.59%, and the partial productivity of N and P by 2.64% and 2.63%, respectively, compared with the OF treatment. However, fertilization did not significantly affect the average yield, harvest indexes of N and P, and partial productivity of N and P. The contents of TN, AN, SOM, TP, AP, and AK across all the treatments decreased significantly with increasing soil depth, while soil pH increased with soil depth. The SF treatment could more effectively increase soil pH and NH_4_^+^-N content compared to the SRF and WSF treatments, while the SRF treatment could greatly enhance other soil nutrients and enzyme activities compared to the SF and WSF treatments. A correlation analysis showed that rice yield was significantly positively associated with tiller number, leaf area index, chlorophyll, soil NO_3_^−^-N, NH_4_^+^-N, SOM, TP, AK, and soil enzyme activity. The experimental results indicate that SRF was the best fertilization method to improve rice growth and yield and enhance soil properties, followed by the SF, WSF, and OF treatments. Hence, the results provide useful information for better fertilization management in the Chaohu Lake region of China.

## 1. Introduction

Rice is the world’s main food crop, dominating food production and consumption [1]. Paddy soil in China accounts for about 23% of the worldwide paddy area, and approximately 60% of paddy soil in China covers in the subtropical region [2,3]. One of the dominant cultivation patterns is rape and rice rotation in the Chaohu Lake region of China. With the increasing demand for agricultural products and the growing public awareness of environmental protection, people are adopting organic farming systems instead of traditional ones [4]. At present, although the massive application of chemical fertilizers on conventional paddy soil has achieved high yields in China [5,6], it has also caused a series of environmental problems, such as severe surface water eutrophication, groundwater pollution, emission of ammonia and greenhouse gases, soil quality degradation, and rapid fertility decline [7]. Organic rice production is more effective in reducing adverse environmental effects than conventional rice production. Based on soil testing and fertilizer field trials to determine the amount, period of use, and working methods of soil nutrients, soil-testing formula fertilization is widely promoted in China [8,9,10].

On the other hand, China is rich in organic fertilizer sources, with an annual output of about 1 billion tons of straw (air-dried) and a total of about 20 million tons of total nutrients; the amount of fecal and urine material is about 4.6 billion t (fresh), and the amount of nutrient resources is about 50 million t [11]. In addition, straw return also has many advantages in improving farmland soil quality [12,13], realizing the reutilization of agricultural waste, reducing pollutant emission from straw combustion, and providing environmental benefits to some extent [14,15]. Additionally, slow-release fertilizers control the granular water solubility by slowing the hydrolysis of water-soluble fertilizers since they have semi-permeable layers of different essential oils and secondary and primary nutrients [16]. Chemical synthetic slow-release fertilizers form microsoluble material after the polymerization of traditional chemical fertilizers to slow down nutrient loss, releasing nutrients after degradation under the action of microorganisms [17]. Slow-release fertilizers need only one-off fertilization and can be more labor- and time-saving than conventional chemical fertilizers that need split fertilization.

Moreover, water-soluble fertilizers containing aminoacids are hydrolysates obtained from materials extracted from septic tanks, contributing to better seed germination and inducing plant biological activity. Straw returning and new fertilizers are safe for the environment and contribute to sustainable, high-output, and low-input crop production. Therefore, reducing the use of chemical fertilizers and applying straw returning and new fertilizer can maximize crop yields and alleviate the adverse effects caused by intensive cultivation, which are essential agricultural management measures in current and future agricultural development [18].

Recently, the country and society have paid more and more attention to combining organic materials or slow-release fertilizers to replace some chemical fertilizers in the agricultural sector. So far, various N-saving application patterns have been developed (balanced N fertilization, integrated N management, use of slow/controlled-release fertilizers and water-soluble fertilizers, etc.) [19]. Previous research shows that slow-release fertilizers, water-soluble fertilizers, crop straw, and other organic fertilizers are common fertilizer sources to replace chemical fertilizers in crop production, improving the quality of cultivated land while ensuring the quality of agricultural products and the safety of the ecological environment. Abid et al. reported that 40% chemical N replaced by pig manure promoted higher rice production, followed in descending order by 40% chemical N replaced by organic slow-release farmer fertilizer practices [20]. In both organic (farmyard manure, cow dung, biogas slurry, sugarcane bagasse) and inorganic (gypsum and lignite) amendments, cow dung significantly improved rice growth and physiological parameters [21]. Moe et al. [22] investigated the effects of integrated organic (poultry manure, cow manure, and plant compost) and inorganic fertilizers on the yield and growth parameters of rice varieties; they found that 50% chemical fertilizer + 50% poultry manure resulted in higher rice growth parameters, yield, and yield components than 100% chemical fertilizer, while the yields of 50% CF + 50% cow manure and 50% CF + 50% compost treatments were similar to 100% chemical fertilizer. In this light, previous studies have demonstrated that replacing chemical fertilizer with 30–50% pig manure, cow dung, and poultry manure significantly improved growth parameters and increased rice yields. However, whether soil properties at different depths were regulated after straw was returned and new fertilizers were added to rice planting in Chaohu Lake and a direct or indirect link between them and rice growth and yield remains unclear.

Therefore, the objectives of this study were to (1) evaluate the effects of straw returning and new fertilizer (slow-release fertilizer and water-soluble fertilizer) substitution on rice growth parameters, yield, nitrogen, and phosphorus uptake; (2) study the soil properties in the three depths (0–30 cm, 30–60 cm, and 60–90 cm) under different fertilizers treatments at different growth stages of rice; and (3) derive management options for an efficient fertilization model for the rice in the Chaohu lake region of China.

## 2. Results

### 2.1. Rice Growth Parameters

Straw returning, slow-release fertilizer, and water-soluble fertilizer did have a specific impact on plant height, main root length, tiller number, leaf area index, chlorophyll content, and aboveground dry (AGD) weight (Figure 1 and Appendix A). With the growth process moving forward, the plant height of rice increased, while the main root length increased first and then decreased. The plant height increased most rapidly during the mid-tilling and jointing stage in all treatments (Figure 1a). Fertilizer treatments positively affected rice growth in different growth stages of rice compared with CK treatment. It could be seen that fertilization had a certain synergistic effect on rice growth. The plant height under SF, SRF, and WSF treatments was taller than that of OF treatment, increasing 0.77–5.03%, 2.95–12.13%, and 0.73–6.09%, respectively, while there was little statistically significant difference among different fertilization treatments.

The main root length of rice with SF treatment was longer in the early growth stage, while the WSF treatment was longer in the late stage (Figure 1b). Compared with CK, the main root length was significantly increased at the tillering stage, jointing stage, and heading stage (*p* < 0.05). Compared with OF treatment, the main root lengths for the SF, SRF, and WSF treatment in each growth period increased on average by 0.99–10.40, 0.56–5.38 cm, and 0.34–14.66 cm, respectively. The average main root length of each rice plant in each treatment reached the maximum at the jointing stage. At the tillering stage and jointing stage, the SF treatment was more beneficial to the growth of the main root length, followed by the WSF and SRF treatment, respectively. At the heading stage, the WSF treatment had the longest main root length, followed by the SRF and SF treatment. However, there was no significant difference in main root length among different treatments at the maturity stage (*p* > 0.05). In the early stage of rice growth, straw returning combined with fertilizer was beneficial to improve root binding and promote root longitudinal growth. However, the difference in main root length between different fertilization treatments decreased gradually with the progress of the growth period. In summary, straw returning to the field, slow-release fertilizer, and water-soluble fertilizer combined with fertilizer can promote the growth of crop roots, thus promoting the absorption and utilization of deep soil water by crops and promoting root growth.

The effective tillering number affects the final panicle number, which mainly affects the yield. In this study, compared with the CK treatment, the tiller number demonstrated a certain significance between all the fertilizer controls at different growth stages (Figure 1c). However, there was no significant difference between different fertilization treatments in all growth stages. The largest tiller number was observed at the jointing stage in the SRF treatment. The average tiller number for the SRF treatment was 4.88–13.79% higher than in the OF treatment. Additionally, the tiller number in the SF and WSF treatments did not increase significantly. As shown in Figure 1d, it could be seen that the leaf area index (LAI) in all treatments showed an increasing trend as rice grew. The LAI increased sharply, but it was slightly delayed at the heading stage. The average LAI increase values for the SF, SRF, and WSF treatments were 4.46%, 10.38%, and 2.80%, respectively, compared with OF. Overall, fertilization treatment had a positive impact on LAI, resulting in an increment of 35.02% on average compared to the CK treatment.

Different fertilization treatments had significant effects on chlorophyll content (SPAD) and the aboveground dry weight of rice. SPAD is a dimensionless value indicating the relative value of chlorophyll content in plant leaves, and the larger the SPAD, the higher the chlorophyll content. From the mid-tillering stage to the maturity stage, the SPAD of rice under each fertilization treatment generally decreased, while the aboveground dry weight increased as a whole. The average SPAD increases for SF, SRF, and WSF treatments were 3.36% and 25.58% compared with OF and CK, respectively. The SF treatment resulted in the highest SPAD at each rice growth stage during observation. The aboveground dry weight of the rice fertilized plots was significantly higher than the CK plots. Compared with the OF treatment, the aboveground dry weight associated with the SF, SRF, and WSF treatments increased by an average of 21.41%, 24.33%, and 15.53%, respectively.

### 2.2. Rice Yield and Nitrogen and Phosphorus Uptake

There are significant differences in rice yield and N and P uptake among different treatments (Figure 2 and Table 1). The average grain yield ranged from 8.97 to 11.89 Mg ha^−1^ for rice. The SF, SRF, and WSF treatments increased rice grain yield by 30.65–32.51%, 0.24–1.66% compared to the CK and OF treatments, respectively. Therefore, straw returning and application of slow-release fertilizer and water-soluble fertilizer combined with fertilizer had significant positive effects on rice yield. However, there was no significant difference in yield among different fertilization treatments (*p* > 0.05). The yields were in the sequence SRF > SF > WSF > OF > CK. The yield under the SRF treatment was higher than those under the SF, WSF, and OF treatments.

Additionally, average grain N uptake ranges from 58.78 to 173.93 kg N ha^−1^, and 57.39 to 126.73 kg N ha^−1^ for straw N uptake. Compared with CK, the average N harvest index values for OF, SF, RSF, and WSF increased by 13.91%, 13.25%, 15.24%, and 14.58%, respectively. The SRF and WSF treatments had greater N harvest index values than OF, but the SF treatment had a lower N harvest index. As for the Partial N productivity, SRF treatment was the highest, followed by WSF, SF, and OF. Average grain P uptake ranges from 18.36 to 45.14 kg P ha^−1^ and 21.30 to 43.35 kg P ha^−1^ for straw P uptake, respectively. Compared with CK, the average P harvest index for OF, SF, RSF, and WSF increased 9.37%, 10.79%, 10.08%, and 10.08%, respectively. Contrary to the N harvest index, the P harvest index of SF treatment improved the most compared with OF treatment. For the Partial P productivity, SRF treatment was also the highest, followed by WSF, SF, and OF. In this light, straw returning to the field, application of slow-release fertilizer, and water-soluble fertilizer can sustain crop productivity comparably to the OF treatment. Nonetheless, straw returning combined with fertilizer improved the utilization efficiency of phosphorus fertilizer in rice compared with the other fertilization treatments.

### 2.3. Soil Chemical Properties

The soil pH value showed a steady uptrend during the rice growth stage among the fertilization treatments (Figure 3 and Appendix A). OF treatment at the maturity stage had the highest pH among all the treatments. The mean values of pH under SF (6.66), SRF (6.62), and WSF (6.62) treatment were lower than the CK treatment, decreased by approximately 0.01, 0.14, 0.15, respectively; however, they ranged from 6.52 to 6.80, indicating neutral soil. Compared to the CK treatment, the SF, SRF, and WSF treatments reduced the pH by 0.22%, 2.04%, and 2.27%, respectively, while the OF treatment increased the pH by 2.36%. In total, the soil pH at the maturity stage was 0.01–0.47 units higher than that at the early stage. The soil pH under the SF treatment was higher than under the SRF and WSF treatments. Furthermore, a higher soil pH was found when straw was returned with chemical fertilizers for SF, followed by SRF and WSF. At the same time, soil pH increased with soil depth, and the average pH values in the fertilization treatments were 6.57, 6.67, and 6.73 at the depths of 0–30, 30–60, and 60–90 cm.

There were some differences among all the treatments on the soil nitrogen content, and that of the fertilized plots were higher than the CK plots (Figure 4a–d and Appendix A). As the rice advanced from the regreening stage to the maturity stage, soil TN and AN content generally decreased first and then increased, significantly differing among the three soil layers (*p* < 0.05), while soil NO_3_^−^-N and NH_4_^+^-N content showed no obvious tendencies among these fertilizers treatments, and NO_3_^−^-N content was less than NH_4_^+^-N content. The highest values of soil total nitrogen (TN), alkali-hydrolyzable nitrogen (AN), nitrate nitrogen (NO_3_^−^-N), and ammonium nitrogen (NH_4_^+^-N) content were all at the depth of 0–30 cm. Soil TN and NO_3_^−^-N contents were the highest in the SF treatment and SRF treatment at the maturity stage, respectively, while soil AN content was the highest in the SRF treatment at the heading stage, and NH_4_^+^-N content was the highest in SF at the regreening stage. The mean soil TN content under all treatments with the three soil layers fluctuated within 1.35–3.00, 1.08–2.47, 1.11–2.67 g kg^−1^ and 1.04–3.18 g kg^−1^ in the four rice growing seasons, and within 42.53–151.67, 56.13–137.11, 95.90–231.62 mg kg^−1^, and 72.42–196.70 mg kg^−1^ for soil AN content, respectively. Specifically, the soil TN content in the SF treatment was 13.44–71.27% and 0.56–42.32% higher than that in the CK and OF treatments, respectively. The mean soil TN contents under the SRF and WSF treatments were significantly higher than under the CK treatment by 23.02–77.90% and 13.89–94.24%, and higher than that under OF treatment by 4.95–30.08% and 0.34–60.60%, respectively. Taking NO_3_^−^-N and NH_4_^+^-N content into account, it was found that the SRF, SF, and WSF treatments all increased them in topsoils (0–30 cm) in the regreening stage compared with CK and OF (Figure 4c,d). During most rice growing stages, the mean soil NO_3_^−^-N content at 0–30 cm was the highest for the SRF treatment at the maturity stage, while NH_4_^+^-N was the highest for the SF treatment at the regreening stage. Additionally, mean soil NO_3_^−^-N content under the WSF treatment was lower than that in the SF and SRF treatments in the 30–60 and 60–90 cm layers, while soil NH_4_^+^-N content under the WSF treatment all had the lowest value among straw returning and new fertilizer substitution treatments in all soil depths. Overall, NO_3_^−^-N and NH_4_^+^-N content decreased as soil depth increased in the SF, SRF, and WSF treatments.

Compared with the CK treatment, OF, SF, SRF, and WSF significantly increased the contents of soil organic matter (SOM), soil total phosphorus (TP), soil available phosphorus (AP), and soil available potassium (AK) (Appendix A). The combined application of chemical fertilizers with straw returning, slow-release fertilizer, or water-soluble fertilizer evenly increased SOM by 4.42%, 15.27%, and 7.61% and soil TP by 8.19%, 32.09%, and 10.84% in the 0–30 cm soil depth, respectively, compared to the soil indicators of the OF treatment. In addition, the SF, SRF, and WSF treatments resulted in significantly higher soil nutrients (e.g., AP, and AK) than in the CK and OF in all three soil layers. However, there was no significant difference in soil AK content among the SF, SRF, and WSF treatments. Moreover, the soil TP, SOM, AP, and AK contents were also affected by growth stages; soil AP, SOM, and AK exhibited an overall increasing trend, with soil TP first increasing and then decreasing over the rice growth stages. Furthermore, this experiment revealed that chemical fertilizers combined with straw returning, slow-release fertilizer, or water-soluble fertilizer affected SOM, soil TP, AP, and AK at a soil profile of 0–90 cm. The contents of SOM, TP, AP, and AK across all the treatments decreased significantly with increasing soil depth (*p* < 0.05). In the topsoil (0–30 cm), SRF treatment at the maturity stage showed the highest contents of soil SOM, AP, and AK (Figure 5a,c,d), and it was at the heading stage that SRF treatment showed the highest contents of soil TP (Figure 5b). In the 30–60 cm soil layer, the WSF treatment at the maturity stage had relatively higher SOM than the other plots, but the WSF treatment at the heading stage had the highest soil AP. Below this depth, the SRF treatment at the tillering stage had the highest SOM. The SF treatment at the heading stage had the highest soil TP, while the SRF treatment had the highest soil AP at the maturity stage. The SRF treatment at the heading stage had the highest soil AK of the 60–90 cm soil layers.

### 2.4. Soil Enzyme Activities

The soil enzyme activities of soil neutral phosphatase (SNP) and soil sucrase (SSU), as well as those of soil urease (SUR) and soil catalase (SCAT) in the 0–30 cm soil depth varied in all treatments (*p* < 0.05; Figure 6a–d and Appendix A). Compared to the application of chemical fertilizers alone, SNP, SNP, and SCAT activities significantly increased after straw returning, slow-release fertilizer or water-soluble fertilizer, and chemical fertilizer co-fertilization; however, the application of straw returning and water-soluble fertilizer had minimal effect on SUR activity at the regreening stage, and there were no significant differences throughout the mid to late growth period, ranging between 0.12 and 0.18 mg g^−1^ d^−1^. SNP activity was the highest in the SRF (1.83 mg g^−1^ d^−1^) treatment at the heading stage, followed by WSF (1.74 mg g^−1^ d^−1^) and SF (1.71 mg g^−1^ d^−1^) at the heading stage. In addition, SSU activity was the highest in the SRF treatment at the maturity stage (14.27 mg g^−1^ d^−1^), followed by SF (13.19 mg g^−1^ d^−1^) and WSF (12.78 mg g^−1^ d^−1^) at the maturity stage. SUR and SCAT activity were the highest in the SRF treatment at the regreening and tillering stages, ranging between 0.08–0.24 mg g^−1^ d^−1^, and 1.45–2.88 mL g^−1^ 20 min^−1^, respectively. Specifically, the SRF treatment increased SNP, SSU, SUR, and SCAT activity by 52.69–153.79%, 43.74–62.90%, 55.56–184.15%, 27.99–64.07%, respectively, over the growth period compared to the CK treatment. Overall, SNP and SCAT activities changed throughout the growth period of rice, first increasing and then decreasing. Conversely, SUR activity showed an opposite trend, first decreasing and then increasing. In particular, the trend in SSU activity increased during the growth period of rice.

### 2.5. Correlation Analysis between Rice Growth and Soil Properties

We evaluated the correlation between rice yield, growth parameters, and soil properties in the 0–30 cm soil depth by the Pearson correlation and the principal component analysis (Figure 7 and Figure 8). Rice yield was significantly positively associated with tiller number, leaf area index, chlorophyll, soil NO_3_^−^-N, NH_4_^+^-N, SOM, TP, AK, and soil enzyme activities (R^2^ = 0.32–0.69; *p* < 0.05). There was no correlation between rice yield and soil pH. Correlation analysis showed that plant height was significantly positively affected by tiller number, leaf area index, aboveground dry weight, soil pH, TP, AN, AP, AK, NO_3_^−^-N, SOM, and SSU activity, but negatively correlated with root length and chlorophyll (*p* < 0.05; Figure 7). And root length and chlorophyll content were significantly negatively correlated with aboveground dry weight, soil AP, AN, SOM, and SSU activity (*p* < 0.05). Furthermore, tiller number and leaf area index were all positively correlated with aboveground dry weight, soil pH, NO_3_^−^-N, NH_4_^+^-N, SOM, TP, AN, AP, AK, and soil enzyme activities. The aboveground dry weight was significantly positively correlated with soil pH, TN, SOM, AP, AK, NO_3_^—^-N, AN, TP, and SSU activity; however, it was negatively correlated with SCAT activity. Moreover, soil pH was significantly positively correlated with soil TP, AP, AK, AN, and SSU activity. Soil TN was positively correlated with SOM, NO_3_^—^-N, AN, SSU, and SUR activities, whereas soil TP was positively correlated with the soil AN, SNP activity, SSU activity, and rice growth parameters except for root length and chlorophyll content. SOM was significantly correlated with rice yield, growth parameters, and soil properties except for soil pH, TP, SNP, and SCAT activities. Soil enzyme activities were all significantly correlated with rice yield, leaf area index, soil NO_3_^—^-N, and NH_4_^+^-N.

The results of the principal component analysis were shown in Figure 8a,b. Using rice yield and growth indexes as original variables, the principal component analysis method revealed that the first principal component axis (PC1) and the second principal component axis (PC2) explained 93.1% of the variation, the contribution rate of the PC1 was 80.2%, and the contribution rate of the PC2 was 12.9%. In contrast, using rice yield and soil indices as original variables, the results showed that PC1 and PC2 axes explained 83.4% of the variation. On the other hand, rice yield and growth parameters were substantially affected by slow-release fertilizer, straw returning or water-soluble fertilizer, and chemical fertilizers treatments. The contributions between rice yield and growth parameters, and soil properties among all the treatments in this study were all ranked as the following sequence: SRF > SF > WSF > OF > CK treatment in 0–30 cm soil.

## 3. Discussion

### 3.1. Effects of Straw Returning and New Fertilizers Substitution on Rice Growth

Rice plant type traits, including plant height, tiller number, root length, leaf area index, chlorophyll content, and dry matter weight are important factors that determine grain yield [23,24,25]. Among them, plant height and tiller number are regulated by the transcription factor, regulatory genes of plant architecture, many hormones, and external environmental factors [26,27]. In the study, the maximum values of plant height and tiller number were all observed in the SRF treatment, followed by SF, WSF, and OF. This could result from the fact that slow-release fertilizer could effectively control the nitrogen release by changing the water solubility of fertilizer, improving the nitrogen supply capacity of the soil, and making the nutrient in quantity, time and space comply with the demand of rice, to improve the effective tillering and rice growth. However, the straw returning could inhibit rice tillering [28,29], producing excessive reducing substances under anaerobic conditions, engendering a toxic effect on rice growth [30]. Meanwhile, SF treatment benefited the growth of the main root length at the tillering and jointing stages. Our results are similar to those of Wang et al., who reported that oilseed rape straw returning produced some positive impacts on rice roots, which were affected by straw type, straw returning mode, and amount [31]. Moreover, in the early stage of rice transplanting, soil reoxidation potential, total amount of reductive substances, and Fe^2+^ mass fraction under the condition of rape straw returning had significant effects on rice root growth, but the effects gradually weakened with time, and the effects were minimal in the later stage.

LAI is an essential parameter for monitoring the photosynthetic efficiency of rice [32]. A larger LAI is conducive to promoting photosynthesis and producing more photosynthetic products, thus increasing rice yield [33]. In our study, the average LAI increases for SF, SRF, and WSF treatments were 4.46%, 10.38%, and 2.80%, respectively, compared with OF. Hence, this experiment showed that SRF treatment had the maximum LAI, consistent with Shivay et al. [34]. Further, the number of tillers and LAI were positively correlated with nitrogen levels. Nitrogen is also an important component of chlorophyll and photosynthetic related enzymes [35]. SPAD was also closely related to crops’ the photosynthetic rate and nutritional status, which affects photosynthesis [36]. Our study also found that WSF improved the rice leaf chlorophyll content. The average SPAD increase for SF, SRF, and WSF treatments was 3.36% and 25.58%, respectively, compared with OF and CK. Moreover, nitrogen is an essential component of dry matter accumulation in rice. Organic N is primarily transported as amino acids that function as a positive regulator of growth and grain yield in rice [27]. Because the WSF treatment contained amino acids, it exhibited increased filled grain numbers [37]. The study found that protein hydrolysate positively affected rice growth and yield [38]. Moreover, amino acids can represent vital nitrogen (N) sources for crop growth and yield in agricultural soils [39]. Simultaneously, the inhibitory effect of straw returning on the late stage of rice growth was small, so the dry matter accumulation of rice was more significantly promoted. The more dry matter accumulates, the more food production increases. Currently, rice needs to strengthen tiller bud activity at the early stages, and thus, the tillering stage is preceded by more continuous accumulation and consumption of critical nutrition from exogenous fertilizer and microbial activity than the maturity stage [40]. The water-soluble fertilizer was applied at the tillering and panicle stages, respectively. Consequently, it was consistent with the fertilizer needed for rice growth.

### 3.2. Effects of Straw Returning and New Fertilizers Substitution on Rice Yield and Fertilizer Use Efficiency

Crop yield is the direct embodiment of soil productivity and the most potent indicator of the change in soil fertility. N, P, and K are essential for grain growth, and their contents determine rice yield attributes. Consistent with previous studies, we also found that the chemical fertilizers combined with slow-release fertilizer, straw returning, or water-soluble fertilizer improved rice grain yield compared to chemical fertilizers alone treatment, yield increment in SRF treatment was higher than in SF, WSF treatments (Figure 2). Slow-release fertilizer could provide nutrients for crops for a long time and promote the growth and development of crops by reducing the rate of nutrient release and extending fertilizer efficiency. For example, applying slow-release fertilizer could increase rice plant height, stem diameter, leaf area index, and dry matter accumulation, thereby increasing rice yield [41,42,43,44]. In addition, adequate water and heat during the rice season may facilitate the release and transport of SRU nutrients and the decomposition of residues from the previous crop. The rice yield was the highest when the fertilizer reduction was combined with 70% slow-release fertilizer [45]. And oilseed-rape straw in soil has created a better growth environment for rice yield by raising nitrogen, phosphorus, and potassium. However, straw returning does not necessarily increase rice yields [32]. Polthanee et al. [46] found that straw returning did not significantly affect rice yield.

Additionally, the amount of decomposition and nitrogen release of straw was fast before and slow after straw returned to the field. Thus, straw returning can achieve a certain normal effect in the rice production process, but with reasonable nitrogen fertilizer application amount and nitrogen fertilizer operation mode, rice yield increase is more significant. In this study, although the rice yield under fertilization treatments showed no distinct effect, it was slightly higher than that under the OF treatment. OF treatment applying chemical fertilizers based on the soil-testing formula fertilization not only increases the utilization rate of fertilizer and reduces the amount of fertilizer, but also improves the yield and quality of agricultural products [47]. In addition, the law of diminishing returns from 50 years ago also revealed that increasing fertilizer application would not be a good way to increase crop yields [48]. Generally, the improved yield under straw returning and new fertilizer substitution are mainly due to the effective use of available resources. In conclusion, the organic carbon provided by straw incorporation and the available N slowly released by slow-release fertilizer may be conducive to microbial activity, thus accelerating nutrient cycling, promoting plant growth, and ultimately increasing rice yield [49,50]. WSF produced positive effects on plant growth and the availability of water-soluble nutrients, and water-soluble nutrients can increase nutrient utilization and enhance the yield and quality of crops.

Soil nitrogen mineralization is an important way to promote crop nitrogen uptake and ensure soil-effective nitrogen balance in soil. Nitrogen uptake ability is not only affected by soil N supply and transporter activity but also by other factors such as root morphology and architecture, the root-to-shoot ratio, N and carbon availability, and various environmental conditions [51,52,53]. In the present study, SF, SRF, and WSF significantly increased N and P uptakes compared to the CK treatment, which agrees with those results from field studies. Under equal fertilizer N conditions, our results demonstrated that SRF and WSF treatment had more N harvest index, while SF treatment had less N harvest index, as compared to OF (Figure 2). Partial N productivity was calculated as 63.23 ± 1.38, 64.44 ± 2.41 kg kg^−1^, and 63.53 ± 2.97 for SF, SRF, and WSF treatments, respectively. The nitrogen demand for rice from transplanting to the heading stage was high, while that from the heading stage to the maturity stage was low [54]. Applying slow-release fertilizer and water-soluble fertilizer can synchronize nutrient release and supply with crop nutrient uptake, improving crop nitrogen absorption and utilization. Fu et al. [55] showed that applying slow-release fertilizer could increase the nitrogen use efficiency of both early and late rice, but the increment of late rice was greater than that of early rice. Moreover, one basal fertilizer application of slow-release fertilizer is less time-consuming and labor-intensive than urea, which needs to be applied multiple times during the growing season. Water-soluble fertilizers are easily absorbed by rapeseed plants [56]. Straw returning can fix nitrogen through its nitrogen decomposition and increase the fixed amount of exogenous nitrogen, thus increasing the utilization rate of nitrogen fertilizer. Straw returning and new fertilizer substitution are the practical approaches to reduce N fertilizer input while stabilizing or improving rice yield and increasing partial N and P productivity. Here, SRF treatment showed the highest increment (Table 1). The increase in nitrogen absorption and nitrogen use efficiency led to increased rice yield. In general, chemical fertilizers reduction combined with straw-returning, slow-release fertilizer, and water-soluble fertilizer treatments can increase the utilization of agricultural waste resources and stabilize the output. This experiment started during the rice growing season in 2022 and has been carried out for a short time and the effects of straw returning and new fertilizers on rice yield and its mechanism should be explored through multi-year continuous straw returning and new fertilizer substitution experiments.

### 3.3. Effects of Straw Returning and New Fertilizers Substitution on Soil Properties

In the Chaohu Lake region, straw returning, slow-release fertilizer, and water-soluble fertilizer are common agronomic measures in agricultural production. However, changes in soil properties caused by these management measures have not been fully studied. Soil properties are generally considered to be the soil’s capacity to maintain soil quality and promote plant productivity [57,58]. In the study, we analyzed the effects of straw returning and new fertilizers substitution on several soil properties (e.g., pH, TN, AN, NO_3_^−^-N, NH_4_^+^-N, SOM, TP, AP, and AK) at different soil depths during the growth stages of rice. However, soil properties differed substantially among different fertilizer treatments due to large variations in the characteristics of the fertilizers [59]. For example, Ghafoor et al. [60] found that slow-release fertilizer application significantly enhanced soil available N, P, and K contents. The combined application of slow-release fertilizer and chemical fertilizers resulted in higher soil TN, available nitrogen, and SOM content [61].

On the other hand, applying slow-release fertilizer improved soil enzyme activity, thus promoting organic decomposition and biological nitrogen fixation and ultimately improving soil fertility [62,63]. Zhang et al. [64] found that straw returning significantly increased soil TN and SOM in both rice growth stages. Compared with chemical fertilizers with the same nutrient level, liquid residue had a better promotion effect on plant growth and soil quality [56]. Generally, in the current study, as the crop advanced from the regreening stage to the maturity stage, soil pH, TN, AN, SOM, AP, and AK increased under SF, SRF, and WSF treatments, while soil NO_3_^−^-N and NH_4_^+^-N content showed no obvious tendencies among these fertilizer treatments, and NO_3_^−^-N content was less than NH_4_^+^-N content.

In most growth stages of rice, chemical fertilizers showed soil pH decline in the 0–30 cm soil depth by decreasing 0.04–0.35 units compared with CK. Whereas the replacement of straw and new fertilizers increased soil pH value to a certain extent. It may be because the ammonification and nitrification of urea in soil after adding chemical nitrogen fertilizer released a large number of protons (H^+^), which reduced soil pH value. Compared with the single application of chemical fertilizers, replacing straw returning and new fertilizers can improve the nitrogen utilization rate, thus reducing the NO_3_^−^-N and H^+^ contents in soil and increasing soil pH value. Additionally, the degree of soil acidification is complex. Crop growth removes N and increases H^+^ ions, thus accelerating soil acidification. In addition, soil acidification may be related to the parent material of soil formation [65,66]. Research found that organic fertilizer application increased soil pH, while inorganic fertilizer application decreased it [67,68]. The application of slow-release fertilizer affected soil pH, and the increase of pH is conducive to converting nitrogen and promoting nitrogen uptake [69]. In the present study, we found that TN, SOM, TP, AP, and AK contents were both highest in the upper soil depth and the values decreased with soil depth, which was similar to the previous studies [70,71]. Within the 0–90 cm soil layer, pH values were the lowest in the 0–30 cm soil depth and highest in the lowest measured soil depth (60–90 cm) across all treatments. Although average pH values in the treatment of SF, SRF, and WSF were lower than that of CK, the differences were insignificant. This result was probably because the soil in the paddy field was usually flooded, which prevented nitrification, and nitrification is an acid-producing process that proceeds via the oxidation of NH_4_^+^-N to NO_3_^−^-N [64]. Meanwhile, in the 0–30 cm soil depth, chemical fertilizers combined with straw returning at the maturity stage resulted in higher TN concentrations than the other treatments (Figure 4a). This may be attributed to the mineralization of biomass of crop residues in the soil. The contents of nitrate and ammonium nitrogen were mainly related to soil aeration conditions and agricultural nitrogen application. The paddy soil has been anaerobic under a long-term flooded anaerobic environment, and nitrification is inhibited, and denitrification is active. NO_3_^−^-N in the soil is reduced to N_2_O and N_2_, thus reducing the accumulation of NO_3_^−^-N. In addition, the annual application of nitrogen fertilizer in the paddy field resulted in the gradual accumulation of ammonium nitrogen content, which made the soil NH_4_^+^-N content significantly higher than NO_3_^−^-N. Our study confirmed that the partial substitution of chemical fertilizers with straw returning and new fertilizers significantly increased NH_4_^+^-N and NO_3_^−^-N contents compared with CK treatment, while there was no significant impact compared with optimized fertilization treatment. Fang et al. indicated that optimized fertilization could achieve a balanced supply of various nutrients and meet the needs of crops, which was the key to increasing crop yield and reducing nitrogen loss [72]. Because the nitrate nitrogen was free in the soil solution, it was easily absorbed by plants and also easily lost [73]. The NO_3_^−^-N content only in the 0–30 cm soil layer under SRF treatment was higher than in the OF treatment. This may be because the nitrogen supply of slow-release fertilizer is based on the needs of the crop and the maintenance of more mineral nitrogen in the top soil. NH_4_^+^-N occupied 50.36–67.13% of total inorganic N in both treatments. Furthermore, the content of soil’s available elements is mainly regulated by water content, pH value and total nitrogen. Crops grown in soil can directly absorb and utilize soil AP [74]. Our research found that soil AP content under SF, SRF, and WSF treatments increased compared with OF (1.72–31.05%, 5.36–71.35%, 1.39–70.04% increase, respectively) (Figure 5c). Application of organic fertilizer could increase soil AP content in the previous study [75]. This study also showed that there was no significant effect on soil AK content between different fertilization treatments. We attributed the results to the large potassium storage in soil [76]. However, compared with OF treatment, soil AK under SF, SRF, and WSF treatments averagely increased by 6.16%, 7.62%, and 3.79%, respectively. Consequently, soil chemical properties interact with each other, affecting soil fertility.

Soil enzymes are the main participants in soil substance circulation and energy flow, which can promote the mineralization and decomposition of soil organic matter and the circulation and transformation of soil nutrients. Soil enzyme activity reflects the active degree of soil biochemical reaction and the nutrient cycling status of N, P, and K in soil [77,78]. Enzyme activity is affected by many biological and abiotic factors, such as soil physical properties, soil nutrients, soil microorganisms, human factors, and plant roots. Muarry et al. [79] believed that phenolic acids in root exudates could reduce the utilization of growth media by microorganisms, thus decreasing soil enzyme activity. Fertilizer application changes soil nutrient status and improves crop growth’s nutrient conditions, affecting soil enzyme activities. The enzyme activities and available nutrients in soil increased significantly to some extent with the increase of liquid residue [56]. However, the overuse of nitrogen fertilizers has not only reduced soil fertility, but also has led to a decrease in enzyme activities due to the acidifying effect of N fertilizer application. As our study showed, soil enzymatic activities (e.g., SNP, SSU, SUR, and SCAT) are elevated under SF, SRF, and WSF treatments. The highest activity of all soil enzymes was SRF treatment, increasing SNP, SSU, SUR, and SCAT activity by 52.69–153.79%, 43.74–62.90%, 55.56–184.15%, 27.99–64.07%, respectively, over the growth period compared to the CK treatment. SNP and SCAT activities were found to be minimal at the maturity stage of rice. That may be due to the lower active root mass present in the soil in the maturity stage of the crop [80]. Additionally, SNP and SCAT activities changed throughout the growth period of rice, first increasing and then decreasing. This was attributed to the differences in root exudates at different growth stages of rice. A high concentration of root exudates and rapid growth of plants would stimulate rhizosphere microbial activity. When rice growth was at its peak, rice roots competed with rhizosphere soil microorganisms for nutrients, decreasing rhizosphere soil microbial activity. In addition, SUR and SSU activities were all positively correlated with soil organic matter content and total nitrogen content. In general, the higher the soil fertility, the higher the activity of SSU activity. The correlation analysis also revealed that rice yield was positively correlated with soil chemical properties and soil enzyme activities, indicating that soil chemical properties and soil enzyme activities could reflect soil fertility and quality to a certain extent.

## 4. Materials and Methods

### 4.1. Experimental Site

The experimental site is located in Tongyang Town (31°41′24″ N, 117°37′12″ E), Anhui Province, China. The area is characterized by a subtropical humid monsoon climate, with an average temperature of 15.7 °C and an annual rainfall of 1215 mm. The soil type is paddy soil. The Initial soil physical and chemical properties are shown in Table 2. The daily average temperature and rainfall during the growing period of rice in the study area are shown in Figure 9.

### 4.2. Materials and Experimental Design

The experimental rice variety is Ganyou 735. Seeds were sown on 6 May 2022 and seedlings were transplanted on 20 June 2022, and harvested on 20 September 2022. The experimental straw was an oilseed rape straw, and the total nitrogen, total phosphorus, and total potassium of the oilseed rape straw were 6.65 g kg^−1^, 0.67 g kg^−1^, and 1.76 g kg^−1^, respectively. Two new high-efficiency fertilizers used were slow-release fertilizer (22% N–8% P_2_O_5_–16% K_2_O) (Zhongyan Anhui Hongsifang Fertilizer Co., Ltd., Hefei, China) and water-soluble fertilizer (Shanxi Zhongnong Guosheng Biotechnology Co., Ltd., Taiyuan, China). Moreover, The TN, TP, and TK of water-soluble fertilizer applied at the tillering stage were measured as 80 g kg^−1^, 0 g kg^−1^, and 55.7 g kg^−1^, respectively. While that of water-soluble fertilizer applied at the panicle stage were measured as 3.4 g kg^−1^, 0.3 g kg^−1^, and 0.3 g kg^−1^, respectively.

This experiment was set up with five treatments and three replicates, using a completely randomized block design. Five treatments included: (1) CK: blank control, no fertilizer; (2) OF: optimized fertilizer, chemical fertilizers based on the medium yield recommended fertilizer amount by testing the soil of formulated fertilization, here is 184.5 kg N ha^−1^, 57.75 kg P_2_O_5_ ha^−1^, and 94.50 kg K_2_O ha^−1^; (3) SF: 4.50 Mg ha^−1^ straw returning + chemical fertilizers; (4) SRF: 0.59 Mg ha^−1^ slow-release fertilizer + chemical fertilizers; (5) WSF: 0.60 Mg ha^−1^ water-soluble fertilizer + chemical fertilizers. The OF, SF, SRF, and WSF treatments received the same amount of N, P, and K fertilizer. The details of the experimental treatment fertilizer application rates are listed in Appendix A.

The insufficient nitrogen, phosphorus, and potassium nutrients in SF, SRF, and WSF were supplemented with urea (containing 46% N), superphosphate (containing 12% P_2_O_5_), and potassium chloride (containing 60% K_2_O) in turn. Urea applications were provided in three split doses: 55% as basal, 20% at tillering, and 25% at panicle initiation. Both slow-release fertilizer and phosphate fertilizer were applied once before rice transplanting. The water-soluble fertilizer was applied twice in equal amounts at the tillering stage with flushing and at the panicle stage with spraying. The 60% of potassium fertilizer was applied as basal dressing before rice transplanting, and the remaining 40% was applied at the panicle initiation stage. The experiment was under natural rainfall. Flooding was needed for the rice season to be maintained at a depth of 3–5 cm in the field, except for mid-season aeration and harvesting period. Field irrigation and disease, insect, and grass damage management are carried out following local conventional ways.

### 4.3. Soil and Plant Sampling and Analysis

#### 4.3.1. Soil Sampling and Analysis

The soil was made up of five soil samples collected across each plot from each replication in the 0–30, 30–60, and 60–90 cm depths during the main growth period of rice in 2022. Soil samples were collected on 25 June, 12 July, 24 August, and 30 September. The samples were divided into two parts. One part of the soil sample was air-dried for the determination of chemical properties and enzyme activities and the other part was frozen immediately for inorganic nitrogen analysis. The soil pH (soil-to-water proportion of 1:2.5) was measured by using a calibrated pH meter [81]. Soil total nitrogen (TN), alkali-hydrolyzable nitrogen (AN), nitrate nitrogen (NO_3_^−^-N), ammonium nitrogen (NH_4_^+^-N), soil organic matter (SOM), total phosphorus (TP), available phosphorus (AP) and available potassium (AK) were determined by Bosten’s routine method of Soil Agrochemical Analysis [82]. Soil TN content was measured using an automatic Kjeldahl nitrogen analyzer (8400; Foss Co., Ltd., Hillerød, Denmark). Soil TP content was measured with an ultraviolet and visible spectrophotometer (T6; Puxi Co., Ltd., Beijing, China) using the molybdenum blue method. Soil alkali-hydrolyzable nitrogen (AN) content was determined using the diffusion absorption method. Soil AP content was extracted using 0.5 mol L^−1^ NaHCO_3_ solution and then measured using the molybdenum antimony-D-isoascorbic acid colorimetry method. Soil AK content was determined using the same flame photometry technique. Soil catalase activity (SCAT), soil neutral phosphatase activity (SNP), soil urease activity (SUR), and soil sucrase activity (SSU) were determined by potassium permanganate titration, benzene bisphosphate, sodium phenol hypochlorite, and 3,5-dinitrolic salicylic acid [83].

#### 4.3.2. Plant Sampling and Analysis

Representative rice samples (5 plants) in each plot were selected to investigate the rice growth characteristics at the mid-tillering, jointing, heading, and maturity stages, respectively. Rice plants were manually harvested, dried and weighed. The chlorophyll meter SPAD-502 (Minolta, Osaka, Japan) was used to determine the relative chlorophyll content (SPAD) of the top 3rd leaf of the main stem [84]. SPAD values were measured three times at the upper, middle, and lower positions and the average value was calculated. At the time of harvest, the entire aboveground plant biomass in the plots was sampled and carefully separated from grain, and remaining aboveground biomass before finally oven dried at 75 °C to a constant weight. Nutrient concentration in the grain and straw was determined by the Nesslerization method for N, and the Vanadium molybdate blue colorimetric method for P using the spectrophotometercro [82].

### 4.4. Data Processing and Analysis

Partial N (P) productivity was calculated by the ratio of grain yield to N (P) rate applied [39]. The N (P) harvest index was determined by dividing total grain N (P) accumulation of the grain by total N (P) accumulation of the aboveground part [85]. The amount of nutrients accumulated from the crop was calculated by the following equation [86]:(1)NCR=NAG+NAS=(NCG%×DWG/100)+(NCS%×DWS/100)
where NCR is the total nutrient accumulation (mg plant^−1^), *NAG* is the nutrient accumulation in grain (mg plant^−1^), *NAS* is the nutrient accumulation in straw (mg plant^−1^), *NCG* is the nutrient content in the grain (%), *DWG* is the dry weight of the grain (g), *NCS* is the nutrient content in the straw (%), *DWS* is the dry weight of the straw (g).

### 4.5. Statistical Analysis

Multivariate analysis of variance (ANOVA) was performed following the general linear model univariate procedure using IBM-SPSS Statistics 23 software package (SPSS, Chicago, IL, USA). The effects of treatments and growth stage on rice grain growth, yield, and soil properties were analyzed using the two-way analysis of variance (ANOVA). In each case, The values of different treatments were compared using the Least Significant Difference test (*p* < 0.05), and the results were expressed as mean ± standard error (Mean ± SE). Graphs were constructed with Origin 2023 (Systat Software, Inc., Washington, DC, USA) software.

## 5. Conclusions

This study investigated the effects of straw returning and new fertilizer substitution on rice growth, yield, and soil properties in the Chaohu Lake region of China. The results show that SRF treatment resulted in higher plant height, tiller number, leaf area index and chlorophyll content and increased dry matter accumulation, thus increasing rice yield. Soil pH, AP, SOM, AK content, and SSU activity exhibited an overall increasing trend; soil TN, AN content and SUR activity generally decreased first and then increased; and soil TP, SNP, and SCAT activities first increased and then decreased over the rice growth stages. Meanwhile, the contents of TN, AN, SOM, TP, AP, and AK across all the treatments decreased significantly with increasing soil depth, while soil pH increased with soil depth. The combined application of chemical fertilizers with straw returning, slow-release fertilizer, or water-soluble fertilizer generally increased soil pH, TN, SOM, TP, available nitrogen, phosphorus and potassium, and enzyme activities compared to the OF treatment value of soil indicators. Among them, SRF treatment had the best effect on the enhancement of soil nutrients and enzyme activities. A correlation analysis showed that rice yield was positively associated with tiller number, leaf area index, chlorophyll, soil NO_3_^−^-N, NH_4_^+^-N, SOM, TP, AK, and soil enzyme activities. Therefore, from the perspective of increasing yield and soil fertility, the alternative treatment of slow-release fertilizer is conducive to improving soil quality and soil fertility and is the key fertilization technology to achieve high rice yield in the Chaohu Lake region of China. Further investigations are required to analyze how the effects of straw returning and new fertilizer substitution in the long-term monitoring. Also, studies regarding the impact of the different fertilization treatments on soil microbial ecology are much needed.

## Figures and Tables

**Figure 1 plants-13-00444-f001:**
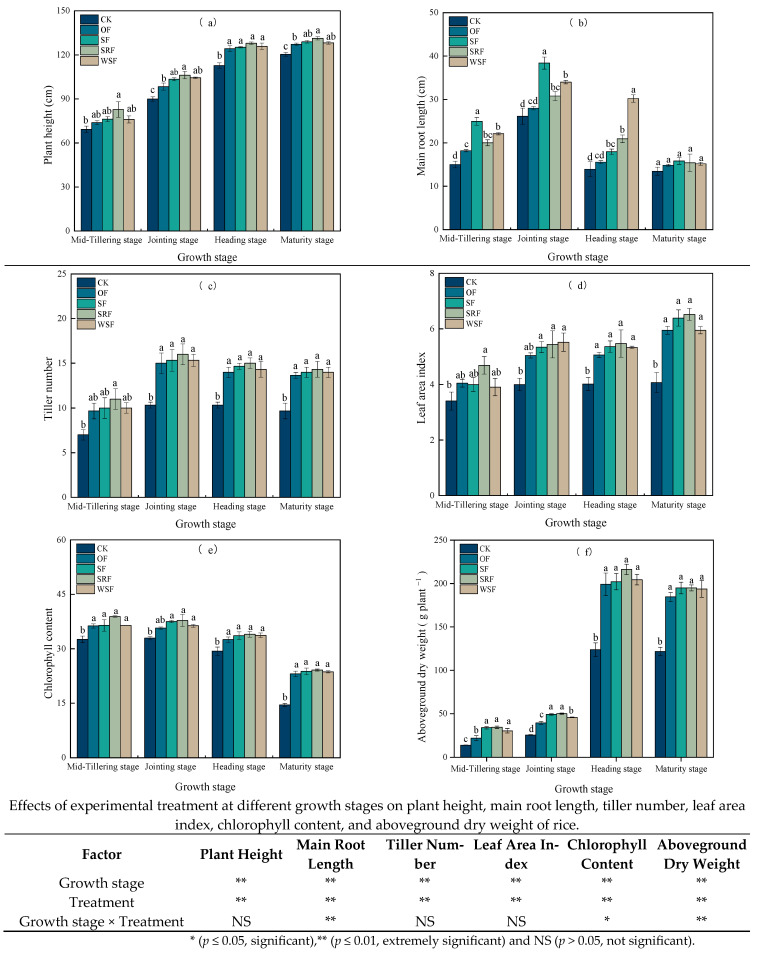
Changes in plant height (**a**), main root length (**b**), tiller number (**c**), leaf area index (**d**), chlorophyll content (**e**), and aboveground dry weight (**f**) during observation. CK: no fertilization; OF: chemical fertilizers based on the medium yield recommended fertilizer amount by testing the soil of formulated fertilization; SF: 4.50 Mg ha^−1^ straw returning + chemical fertilizers; SRF: 0.59 Mg ha^−1^ slow-release fertilizer + chemical fertilizers; WSF: 0.60 Mg ha^−1^ water-soluble fertilizer + chemical fertilizers. Different letters indicating significant differences based on LSD (*p* < 0.05).

**Figure 2 plants-13-00444-f002:**
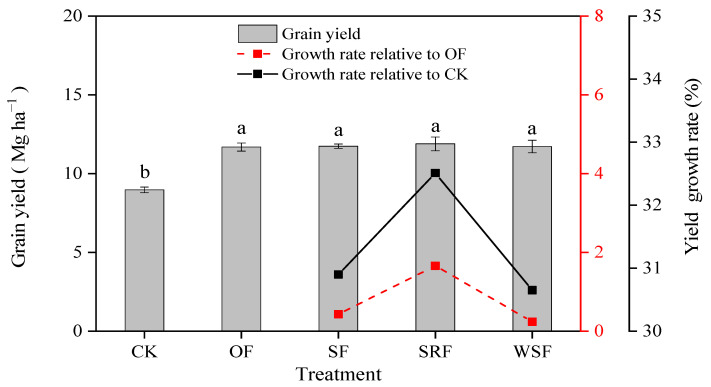
Effects of different fertilizer types on rice yield. Different lowercase letters indicate significant differences between different treatments (*p* < 0.05).

**Figure 3 plants-13-00444-f003:**
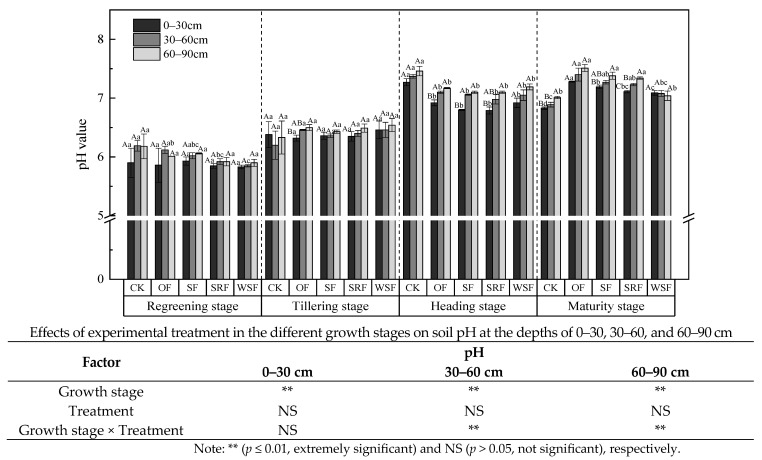
Soil pH value in the regreening, tillering, heading, and maturity stages under five treatments. Different capital letters indicate significant differences at the different soil depths in the same treatment (*p* < 0.05). Different lowercase letters indicate significant differences between different treatments at the same depth (*p* < 0.05).

**Figure 4 plants-13-00444-f004:**
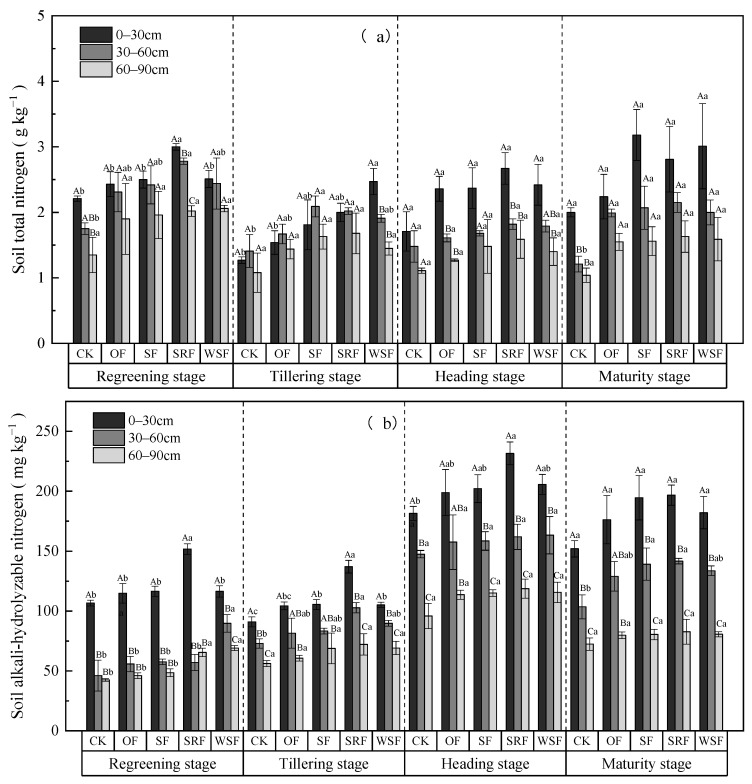
Changes in soil total nitrogen (**a**), alkali-hydrolyzable nitrogen (**b**), nitrate nitrogen (**c**), and ammonium nitrogen (**d**) during observation. Different capital letters indicate significant differences at the different soil depths in the same treatment (*p* < 0.05). Different lowercase letters indicate significant differences between different treatments at the same depth (*p* < 0.05).

**Figure 5 plants-13-00444-f005:**
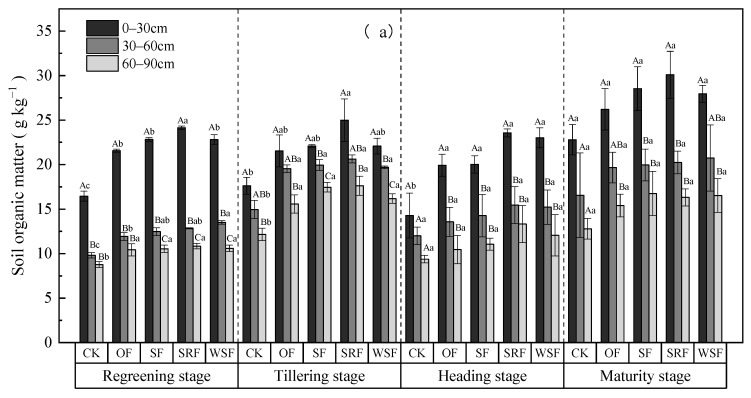
Changes in soil organic matter (**a**), soil total phosphorus (**b**), soil available phosphorus (**c**), and soil available potassium (**d**) during observation. Different capital letters indicate significant differences at the different soil depths in the same treatment (*p* < 0.05). Different lowercase letters indicate significant differences between different treatments at the same depth (*p* < 0.05).

**Figure 6 plants-13-00444-f006:**
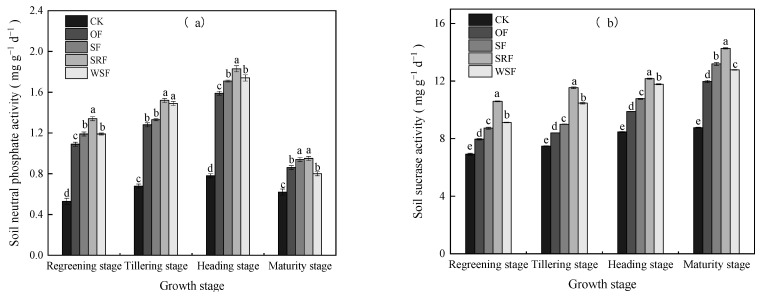
Changes in Soil neutral phosphatase (**a**), Soil sucrase (**b**), Soil urease (**c**), and Soil catalase activity (**d**) in the 0–30 cm soil depth during observation. Different lowercase letters indicate significant differences among the treatments (*p* < 0.05).

**Figure 7 plants-13-00444-f007:**
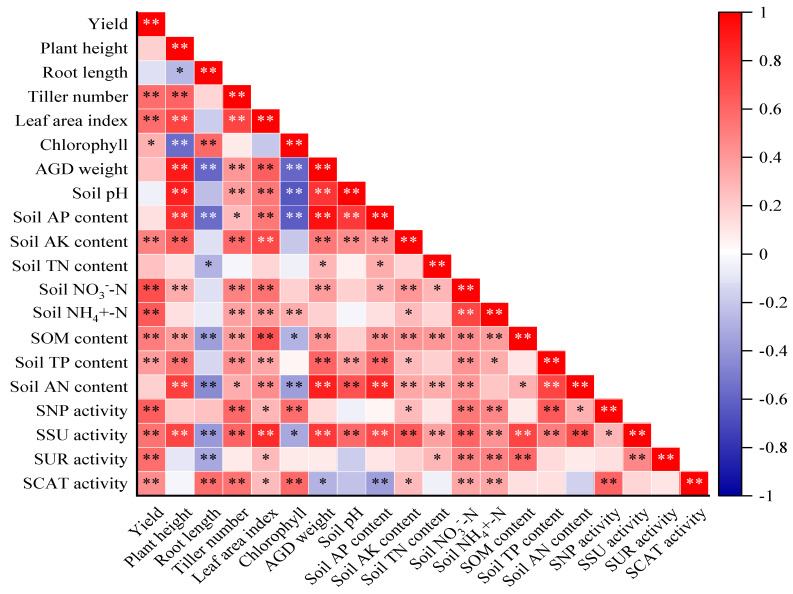
Pearson correlations between rice yield, growth parameters, and soil properties in the 0–30 cm soil depth. Asterisks denote significance at the *p* < 0.05 and *p* < 0.01 probability levels (* and **, respectively). Red indicates a positive correlation; blue indicates a negative correlation.

**Figure 8 plants-13-00444-f008:**
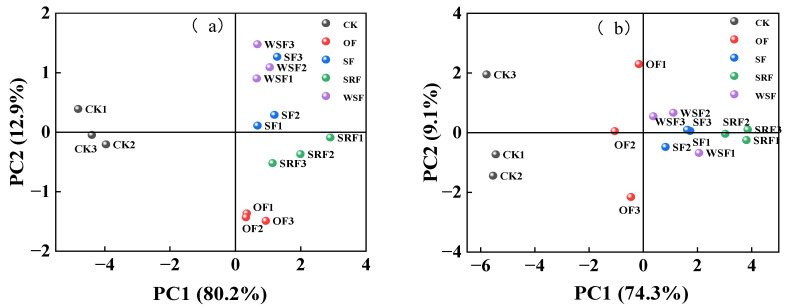
Principal component analysis of rice yield and growth parameters (**a**), soil properties in the 0–30 cm soil depth (**b**) with different fertilizer management.

**Figure 9 plants-13-00444-f009:**
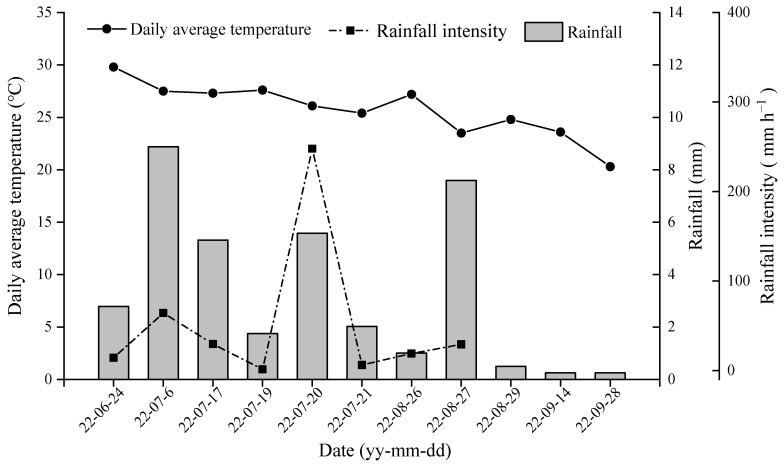
The daily average temperature and rainfall in Tongyang Town.

**Table 1 plants-13-00444-t001:** Grain N (P) uptake, straw N (P) uptake, N (P) harvest index, and partial N (P) productivity in different treatments.

Experimental Treatment	Grain N Uptake (kg ha^−1^)	Straw N Uptake (kg ha^−1^)	N Harvest Index	Partial N Productivity (kg kg^−1^)	Grain P Uptake (kg ha^−1^)	Straw P Uptake (kg ha^−1^)	P Harvest Index	Partial P Productivity (kg kg^−1^)
CK	58.78 ± 3.07 c	57.39 ± 2.18 b	0.50 ± 0.01 b	/	18.36 ± 0.71 c	21.30 ± 0.99 c	0.463 ± 0.01 b	/
OF	144.77 ± 5.37 b	108.35 ± 2.89 a	0.57 ± 0.01 a	62.78 ± 1.58 a	34.91 ± 1.20 b	33.86 ± 1.47 b	0.507 ± 0.00 a	200.58 ± 5.07 a
SF	153.87 ± 2.10 ab	114.64 ± 4.01 a	0.56 ± 0.01 a	63.23 ± 1.38 a	35.60 ± 0.85 b	33.95 ± 0.64 b	0.513 ± 0.00 a	202.02 ± 4.41 a
SRF	173.93 ± 9.86 a	126.73 ± 5.53 a	0.58 ± 0.01 a	64.44 ± 2.41 a	45.14 ± 2.23 a	43.35 ± 1.37 a	0.510 ± 0.01 a	205.87 ± 7.69 a
WSF	149.42 ± 5.53 b	110.59 ± 9.57 a	0.58 ± 0.01 a	63.53 ± 2.97 a	42.78 ± 1.78 a	41.27 ± 2.30 a	0.510 ± 0.01 a	202.98 ± 9.48 a

Note: Data (mean ± standard errors, *n* = 3) with different letters indicate a statistically significant difference based on LSD (*p* < 0.05). The symbols in the following tables and figures are the same as those in this table.

**Table 2 plants-13-00444-t002:** Physical and chemical properties of the experimental soil.

Soil Depth (cm)	BD (g cm^−3^)	pH	TN (g kg^−1^)	TP (g kg^−1^)	AK (mg kg^−1^)	AP (mg kg^−1^)	AN (mg kg^−1^)	SOM (g kg^−1^)
0–30	1.25	7.08	2.09	0.17	130	14.96	91.00	19.54
30–60	1.40	6.87	1.11	0.16	110	5.83	25.48	9.56
60–90	1.47	6.91	1.23	0.17	100	3.82	21.84	6.94

Note: BD: bulk density, TN: total nitrogen, TP: total phosphorus, AK: available potassium, AP: available phosphorus, AN: alkali-hydrolyzable nitrogen, SOM: soil organic matter.

## Data Availability

The data presented in this study are available on request from the corresponding author.

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
