# Peer review of "Effects of Straw Returning and New Fertilizer Substitution on Rice Growth, Yield, and Soil Properties in the Chaohu Lake Region of China"

_plants, 2024, doi:10.3390/plants13030444_

Round 1
Reviewer 1 Report
Comments and Suggestions for Authors
Comments and Suggestions for Authors
The current study entitled “Effects of Straw Returning and New Fertilizers Substitution on Rice Growth, Yield, and Soil Properties in the Chaohu Lake Region of China” investigated the effects of straw returning and new fertilizers on rice growth parameters, soil properties under different fertilizers treatments at different rice growth stages and derive management options for an efficient fertilization model for the rice in the Chaohu lake region of China”.
Overall, the paper has been well written, with impressive results and adequate discussion. However, there are some comments:
The text's English is readable, but has some grammatical mistakes and repetitive words, especially in the introduction. Please check whole the text to avoid any mistakes.
Abstract
The abstract provides a clear objective regarding the study; However, the material only included the treatments, and I would suggest that the authors add some details and key points. Also, the results are very long, and it would be beneficial to modified, because it is abstract, and it is better to bring only significant result.
Introduction
Introduction needs more extent. Background of research is not enough, somewhere, authors tried to refer to some agricultural findings in their background of study (lines 77-87), However, there are very general and not enough, the authors can find good published works with the same idea but different fertilizers, so, it needs to make a good background based on those studies.
In line 49, authors mention that “….massive application of chemical fertilizers on ….”, conventional system of farming based on chemical fertilizer, and since the main aim of this study is the function of new fertilizers substitution in rice farming, I would suggest that the author add one or two sentences about organic and conventional rice farming between lines 45-49. This can a good point of view to readers and improve the novelty of the current work. Here is a published work that you can use it https://doi.org/10.3390/su142315870.
Lines 49-50: In one sentence, there are three fertilizer words! Please paraphrase the sentence.
Lines 57-62: Unclear sentence. Please rewrite it.
Also, in lines 91-93 the authors mention that “Little attention has been paid to the different depths for straw returning and new fertilizers treatments in rice agricultural systems”, this statement is very general, please add the result of that research and highlight the gap of knowledge in previous studies.
Results and discussion
The results are well written, also the Pearson correlations between rice parameters are impressive and complete the results.
The data of the current research is worthwhile for further meta analyses research, so I would suggest that the authors add the data (+SD or SE) in the table form in the supplementary file. It is only a suggestion, and the authors can ignore it.
Material and methods
Please add reference(s) for equation1.
Conclusion
It is better if the authors add suggestions for future research at the end of the conclusion.
Comments on the Quality of English Language
The text's English is readable, but has some grammatical mistakes and repetitive words, especially in the introduction. Minor editing of English language required,
Reviewer 2 Report
Comments and Suggestions for Authors
The manuscript “Effects of Straw Returning and New Fertilizers Substitution on Rice Growth, Yield, and Soil Properties in the Chaohu Lake Region of China” uses the different depths for straw returning and new fertilizers treatments in rice agricultural systems to evaluate their effects on the rice growth parameters, yield, nitrogen, and phosphorus uptakes; and also measured the soil properties in the three depths under different fertilizers treatments at different growth stages of rice, to derive management options for an efficient fertilization model. The results indicated that reduced fertilization combined with slow-release fertilizer could improve rice growth and yield and enhance soil properties more than straw-returning and water-soluble fertilizer, which is useful for rice management. Overall, the manuscript is well structured and the contents are relevant. However, the manuscript could not be considered for publication due to the issues mentioned below.
Abstract
1. Lines 17-18 ‘chemical fertilizers with straw returning treatment (SF), mixed chemical fertilizers and slow-release fertilizer treatment (SRF), mixed chemical fertilizers and water-soluble fertilizer (WSF).’ Abbreviations don’t follow the rules, it is better to change the order of description, for example: change ‘chemical fertilizers with straw returning treatment (SF)’ to ‘straw returning treatment with chemical fertilizers (SF)’.
Result
2. Lines 102, Remove ‘Both’
3. Please review all the text, and remaining units for the unit writing standards. Like ‘g·kg-1’ remove ‘·’ change unit ‘g/kg’ to ‘g kg-1’
4. It would be better to use the Mg ha-1 to replace kg ha-1
5. The arrangement of the graphs needs to be standardized. Fig .4, Fig.5, Fig.6, Fig.8.
6. Which depths of soil were used in the correlation analysis with soil properties in Fig.8?
Materials and methods
7. Line 582 Table, Please remain units for the unite writing standards, such as change the unite ‘g/kg’ to ‘g kg-1’
8. Lines 579-604 also need to be modified for the abbreviation.
9. Line 600-604 How much chemical fertilizer (N? P? K?) was applied in the SF, SRF, and WSF treatments, respectively?
10. Please show detailed information about the sampling date for the soil collection.
11. Please show detailed information about which depth soil was used to analyze the correlation between the soil characteristics and plant parameters.
12. Statistical analysis was shown using one-way ANOVA, does the growth stage have a significant effect on the different parameters??
Discussion
This part also needs to be improved and condensed, it needs more explanation and comprehensive discussion.
Literature
Reduce the number of literature citations.
Comments on the Quality of English Language
Required the editing of the English
